# In Situ Hybridization of Feline Leukemia Virus in a Case of Osteochondromatosis

**DOI:** 10.3390/vetsci9020059

**Published:** 2022-01-31

**Authors:** Anna Szilasi, Zsófia Koltai, Lilla Dénes, Gyula Balka, Míra Mándoki

**Affiliations:** 1Department of Pathology, University of Veterinary Medicine, 1078 Budapest, Hungary; szilasi.anna@univet.hu (A.S.); denes.lilla@univet.hu (L.D.); mandoki.mira@univet.hu (M.M.); 2DUO-VET Veterinary Clinic, 1078 Budapest, Hungary; zskoltai@gmail.com

**Keywords:** osteochondromatosis, FeLV, in situ hybridization, feline

## Abstract

Osteochondromatosis, also known as multiple cartilaginous exostosis, polyostotic osteochondroma, and multiple osteochondromas, comprises one-fifth of all primary bone tumors in cats, with no breed or sex predisposition or hereditary pattern. Unlike in dogs, horses, and humans, it is predominantly seen in young cats (2–4 years old), after the maturation of the skeleton. Although the pathogenesis of osteochondromatosis is not fully understood, it is considered to be related to infection by feline leukemia virus (FeLV) or other retroviruses, such as the feline sarcoma virus. However, the presence of viral particles within tumor lesions has only been demonstrated by electron microscopy. The malignant transformation of osteochondromas, most typically to osteosarcoma or chondrosarcoma, has also been attributed to the viral infection. Here we report the case of osteochondromatosis in a 3.5-year-old male domestic European shorthair cat with concurrent FeLV infection confirmed by polymerase chain reaction. Viral RNA was visualized in representative tissues (spleen, mesenteric lymph node, liver, kidney, lung, brain) and in the osteochondromas with RNAscope in situ hybridization, which supports that FeLV infection may be involved in the pathogenesis of osteochondromatosis.

## 1. Introduction

Osteochondromatosis also known as multiple cartilaginous exostosis, polyostotic osteochondroma, and multiple osteochondromas comprises 20% of all primary bone tumors in cats, with no breed or sex predisposition or hereditary pattern. It is a rare condition in cats, with only a few descriptions and case reports in the available literature [1,2,3,4,5,6]. The term osteochondromatosis refers to a disease in which multiple benign, cartilage-capped tumors arise from bone surfaces as a result of endochondral ossification. Besides cats, the disease has been reported in pigs, horses, dogs, and a macaque [7,8,9,10,11]. The syndrome has shown a familial/hereditary background in horses, dogs, and pigs, similarly to humans, and it occurs typically in young animals during the period of active bone growth [12,13].

Feline osteochondromatosis shows different pathobiology, clinical aspect, and most likely has a different etiology: it is predominantly seen in young cats, where it is most often diagnosed at the age of 2–4 years [2], when the skeleton has already matured. Moreover—with a few exceptions [5,6,14]—feline osteochondromatosis is typically found in cats infected with feline leukemia virus (FeLV) [1,3,4]. Type C retroviral particles resembling FeLV and feline sarcoma virus has been observed by electron microscopy in the cartilage caps of cats with osteochondromatosis [1,15]. Presumably, FeLV infection of periosteal fibroblasts is linked to the special pathogenesis of feline osteochondromatosis [16]. However, the role of FeLV infection in the development of this disease has not been fully elucidated. The viral etiology might explain the different manifestation of the disease in cats compared to other species.

In cats, the disease affects the axial skeleton, most commonly the rib cage, scapulae, vertebral column, skull, pelvis, and limbs. Feline osteochondromatosis tumors show progressive growth and can substantially deteriorate the quality of life of the affected patients. Moreover, malignant transformation to osteosarcoma or chondrosarcoma may occur [6,17]. Marked clinical signs, including lameness and neurologic signs in case of vertebral involvement, may occur in the affected cats. As a treatment, temporary local improvement can be achieved by palliative surgical debulking of the neoplasms, but osteochondromatosis (and the concurrent FeLV infection) cannot be cured, and the condition usually leads to euthanasia [2,17]. The reported survival time is usually less than a year after diagnosis [6].

The pathogenesis of feline osteochondromatosis is not fully understood, and the concurrent FeLV infection in some cases has only been confirmed from peripheral blood by polymerase chain reaction (PCR). Presence of FeLV genome was detected in the tumors by electron microscopy [1,4]. The condition is generally associated with FeLV infection, although the exact role of the infection in the background of the disease is not yet revealed.

Several techniques have already been used for the visualization of FeLV infection in tissue sections, such as electron microscopy, immunohistochemistry, and fluorescent in situ hybridization [18,19,20]. RNAscope in situ hybridization (ISH) is a novel technique based on the detection of RNA molecules in tissue samples [21]. RNAscope has a higher sensitivity compared to classic ISH, which enables the detection of low viral loads with extreme specificity. Moreover, it provides a very accurate localization and visualization of infected cell type.

The primary aim of our study was to describe the case of a 3.5-year-old European shorthair cat affected by osteochondromatosis and detect the presence of FeLV RNA within the tumor tissue and other organs using the novel RNAscope in situ hybridization technique [22].

## 2. Case Presentation

A 3.5-year-old castrated male European shorthair cat was admitted to a veterinary clinic, where a small (1 cm in diameter), firm nodule was noticed behind the left scapula during a complete general physical examination. The nodule was clearly visible with radiographic imaging—a calcified, radiodense, and well-circumscribed lesion could be seen. The cat did not show other clinical symptoms at that time. After eight months, the range of motion of both forelimbs was mildly reduced as several other neoplasms grew at the thoracic vertebral and sternal areas, as well as on the rib cage (both sides). The disease progression was continuous according to the owner. These tumors were dense, calcified, moderately demarcated bony structures according to the X-ray scan, some of them protruded into the thoracic cavity (Figure 1). Laboratory tests, including complete blood count (CBC) and serum biochemical profile, showed mildly decreased total protein level and increased urea and creatinine levels indicating an IRIS2 (International Renal Interest Society) stage of renal insufficiency. Other blood parameters were within the physiological ranges. There were no abnormalities or signs of further metastases during the abdominal ultrasound examination. The known FeLV-positive status of the animal was reconfirmed from peripheral blood by the use of Witness FeLV-FIV test (Zoetis, Parsippany–Troy Hills, NJ, USA). The cat had not been vaccinated against FeLV; it only received regular combined vaccinations (against feline parvovirus, feline calicivirus, and feline herpesvirus 1). After the definite diagnosis of feline osteochondromatosis, due to the poor prognosis, the patient was humanely euthanized.

The carcass of the cat was admitted to the Department of Pathology, University of Veterinary Medicine, Budapest, for necropsy. Gross postmortem examination of the cat showed multiple, large, firm masses (the smallest was 3 cm, the largest was 8 cm in diameter) connected to the sternum, the ribs, and the thoracic vertebrae of the animal (Figure 2). The liver was moderately enlarged, more yellowish than normal, suggesting mild steatohepatopathy. All other organs appeared normal.

### 2.1. Materials and Methods

Representative samples of the tumors and all major organs (spleen, liver, kidney, small and large intestines, mesenteric lymph node, pancreas, adrenal glands, heart, lung, thyroid glands, and brain) were fixed in 10% neutral-buffered formalin for 24 h at room temperature. The osseous neoplasms were decalcified in formic acid solution (37% HCl, 99% HCOOH, H_2_O) for 4 days before further processing. Formalin-fixed and paraffin-embedded sections were stained with hematoxylin and eosin and submitted to routine histological examination.

The presence of FeLV genetic material in sampled tissues was tested by two methods: PCR and RNAscope in situ hybridization. For the end-point polymerase chain reaction, a pool of fresh tissues (mesenteric lymph node, liver, spleen), bone marrow, and the proliferative lesion were examined separately. The organ and tumor samples were homogenized with TissueLyser LT (Qiagen, Hilden, Germany) and after centrifugation, the RNA was isolated from the supernatant in a QIAcube automatic instrument (Qiagen, Hilden, Germany) using QIAmp cador Pathogen Mini Kit (Qiagen, Hilden, Germany) according to the manufacturer’s instructions. The forward 5′-AACAGCAGAAGTTTCAAGGCC-3′ and the reverse primers 5′-TTATAGCAGAAAGCGCGCG-3′ amplified proviral DNA (early-stage products) of the viral reverse transcription process [23]. The reactions were performed using a One-Step RT-PCR Kit (Qiagen, Hilden, Germany) in a Genesy 96T gradient PCR machine (Tianlong, Xi’an, China). A known positive sample was used as a positive control. The reaction mixture without adding any nucleic acid served as a negative control. A master mix containing 5.7 mL RNase-free water, 2 mL 5× buffer (final concentration: 1×), 0.4 mL dNTP Mix (final concentration: 400 µM of each dNTP), 0.4 mL enzyme mix (no concentration is given by the manufacturer), 0.1 mL RNase inhibitor (final concentration: 5 units/reaction), 0.2 mL forward and reverse primers each (40 µM; final concentration: 0.6 µM) was added into tubes with 0.5 mL sample. Thermal conditions were 50 °C for 30 min and 94 °C for 3 min in the stage of reverse transcription and initial denaturation; 95 °C for 15 sec and 60 °C for 1 min and 72 °C for 1 min in stage 2 (45 cycles); 72 °C for 10 min. The PCR reaction was then subjected to agarose gel electrophoresis and the amplicons visualized under UV light.

For the RNAscope ISH assay, 5 µm thin sections were cut and mounted on SuperFrost Plus slides (Thermo Fisher Scientific, Waltham, MA, USA) [21]. Steps of ISH were performed according to the manufacturer’s instructions, manually (with RNAscope 2.5 high definition (HD) red kit; Advanced Cell Diagnostics, Newark, CA, USA) as described earlier [22]. Briefly, sections were deparaffinized in xylene (2 × 5 min), then a series of 96% alcohol was used for rehydration (2 × 3 min). After 10 min of incubation in hydrogen peroxide at room temperature, sections were exposed to a retrieval buffer for 15 min at 95–102 °C and treated with protease at 40 °C for 30 min. The step of hybridization with the target RNA-specific oligonucleotide probes for FeLV (RNAscope Probe-V-FELV-env (catalog no. 491001)), a housekeeping positive control (RNAscope positive control probe Fc-PPIB (catalog No. 455011)), and a bacterial negative control (RNAscope negative control probe-DapB (catalog 310043)) was carried out for 2 h at 40 °C. Signal amplification steps (with amplifiers 1–6) were done after the hybridization according to the manufacturer’s instructions. Labeled probes containing a red chromogenic enzyme bind to multiple binding sites on the amplifiers. Gill II modified hematoxylin treatment for 1 min was used for counterstaining and the slides were cover-slipped after 15–20 min air drying at 60 °C. Each single targeted viral RNA transcript molecule appeared with light microscopy as a distinct dot of red chromogen precipitate.

The slides were scanned with a Pannoramic Midi II slide scanner instrument (3DHistech, Budapest, Hungary), the digital slides were analyzed, and the representative pictures were taken with the SlideViewer software (3DHistech, Budapest, Hungary).

### 2.2. Results

Histopathologic examination of the representative samples obtained from the bony tumors revealed multiple irregular islands of well-differentiated hyaline cartilage surrounded by osteoid tissue and bone trabeculae originating from endochondral ossification (Figure 3). Routine histopathologic examination of sampled organs with hematoxylin and eosin staining revealed mild pathologic lesions only: mild diffuse microvesicular steatohepathopathy in the liver, multifocal nodular hyperplasia in the pancreas, and mild lymphoid depletion and paracortical hyperplasia in the mesenteric lymph node. We did not find any alterations in kidney sections, which could explain the increased urea and creatinine levels in peripheral blood.

Every tissue sample (a pool of organs (mesenteric lymph node, liver, spleen), bone marrow, and the proliferative lesion, examined separately) was positive for FeLV proviral DNA with the PCR method. The RNAscope in situ hybridization also verified the presence of FeLV RNA in every tissue analyzed (liver, spleen, kidney, mesenteric lymph node, lung, brain, osteochondroma). Bone marrow was not examined with the ISH method. Multifocally distributed, individual or coalescing red dots were seen on the sections with different quantities: large amounts of the viral RNA-positive cells (predominantly lymphocytes and other lymphoid cells) were found in the spleen, in the mesenteric lymph node, and in the liver. A lower amount was present in the frontal area of the brain (Figure 4), kidney, and lungs. Although in remarkably lower amounts compared to other organs, FeLV RNA-positive cells were identified sporadically in the decalcified osteochondromatous lesions as well (Figure 5). The positive cells were found in the osseous compartment, most likely within the cytoplasm of osteoblasts. To compare the effect of decalcification on tissues, positive control ISH assay was run on the same decalcified tumor tissue—only a few endothelial cells and cells of connective tissue were identified as positive.

## 3. Discussion

A 3.5-year-old neutered domestic European shorthair cat was euthanized due to the progressive proliferation of large osseous masses connected to the sternum, ribs, and spinal column of the animal. According to patient history, clinical signs, laboratory results, gross and microscopic lesions, and the cat’s FeLV-positive status, the definitive diagnosis of feline osteochondromatosis was established. The case presented is the first to successfully visualize FeLV genome within the osteochondromatosis neoplasms by the use of RNAscope in situ hybridization. Moreover, we could detect the virus in tissues without severe histopathologic lesions, such as liver, spleen, kidney, mesenteric lymph node, lung, and brain. Previous reports verified the presence of FeLV of the affected animals in peripheral blood by PCR mostly [3,4], although within the osteochondromatosis tumors the presence of retrovirus-resembling particles was only seen with electron microscopy [1,4]. No specific detection and visualization of the viral genome were reported so far; moreover, cases of FeLV-negative cats with osteochondromatosis were described earlier [5,6], which raises the question of the contribution of FeLV in the pathomechanism of this condition. The presence of red signal, e.g., the amount of FeLV positive cells, was relatively low in the lesions but very high in spleen, lymph node, and liver. This observation can be explained two ways: (a) there are only a few FeLV target cells in the osteochondromatous tissues, or (b) yet to be identified cells and cell types are infected, but the formic acid solution during the process of decalcification degrades the RNA leading to loss of sensitivity [24]. Further methods and/or other types of decalcifiers should be tested in the future to determine if they have less effect on RNAscope ISH, and thus visualization of viral RNA in osseous lesions. Demonstrability of the viral genome was probably hindered due to formic acid decalcification (and therefore degradation of FeLV RNA), as it was confirmed by the use of a housekeeping positive control probe [23], but this step is inevitable in processing osseous tissues. The use of other decalcifier agents (e.g., EDTA) should be considered in case of possible additional examinations besides routine histopathology.

The presence of retroviral particles in situ within tumors has been only demonstrated by electron microscopy. In one report, immunohistochemistry failed to detect FeLV antigens in tumor cells, when positivity was observed in hematopoietic cells of bone marrow [4]. In this case, with the use of RNAscope ISH assay, FeLV RNA was detected in the tumors as well, although in low amount, which can be a step forward to further clarify the role of FeLV in the etiology of feline osteochondromatosis. RNAscope is a sensitive and specific method to detect RNA, which can be a huge advantage in the examination of infectious diseases or viral oncogenesis. The disadvantage is that this method is yet very expensive, and therefore used mostly for research purposes and not in routine diagnostics.

## Figures and Tables

**Figure 1 vetsci-09-00059-f001:**
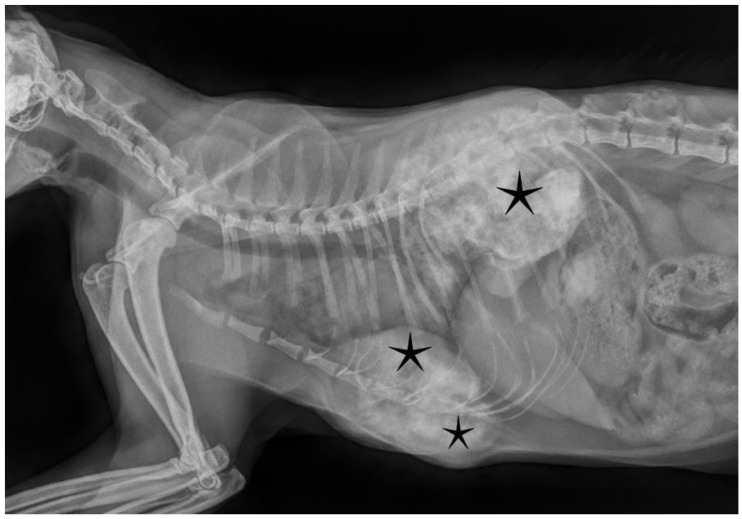
Radiographic imaging showed calcified, moderately demarcated tumorous lesions (asterisks) protruding into the thoracic cavity, deforming normal skeletal structures.

**Figure 2 vetsci-09-00059-f002:**
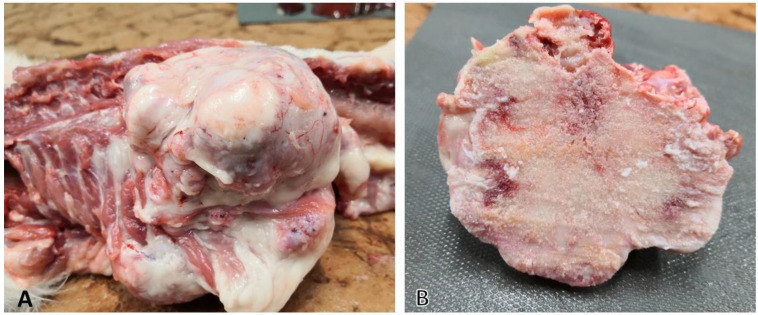
Lesions in osteochondromatosis: (**A**) tumors originating from bones (vertebrae and ribs); (**B**) cut surface of the same lesion showing mostly osseus substance, but also gray-to-tan cartilaginous tissue could be observed.

**Figure 3 vetsci-09-00059-f003:**
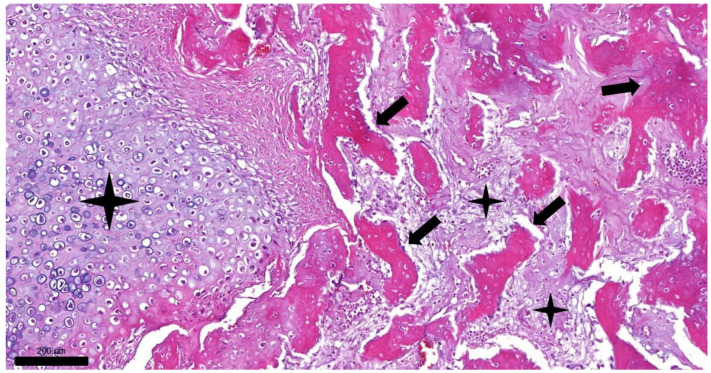
Histopathologic picture of feline osteochondromatosis (hematoxylin and eosin staining, 100× magnification, bar = 200 µm). Trabeculae of bone (arrows) are surrounded by islands of cartilage (asterisks).

**Figure 4 vetsci-09-00059-f004:**
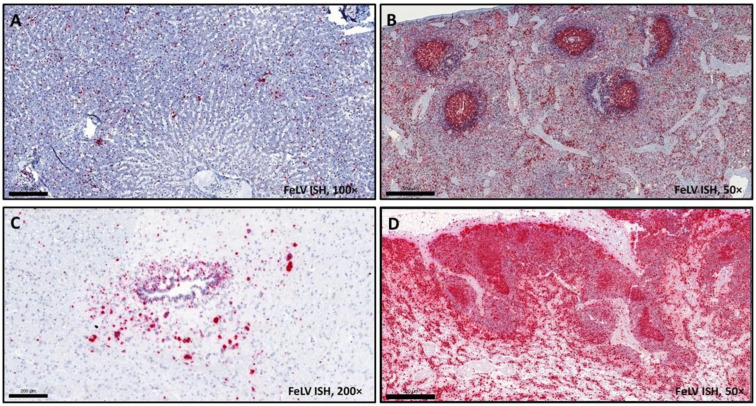
Tissue samples from a cat euthanized due to osteochondromatosis, FeLV ISH (magnification 100× (**A**,**C**), bar = 200 µm; 50× (**B**,**D**); bar = 500 µm). Red chromogen precipitates can be seen in (**A**) liver, (**B**) spleen, (**C**) brain, and (**D**) mesenteric lymph node tissues, in FeLV-infected cells.

**Figure 5 vetsci-09-00059-f005:**
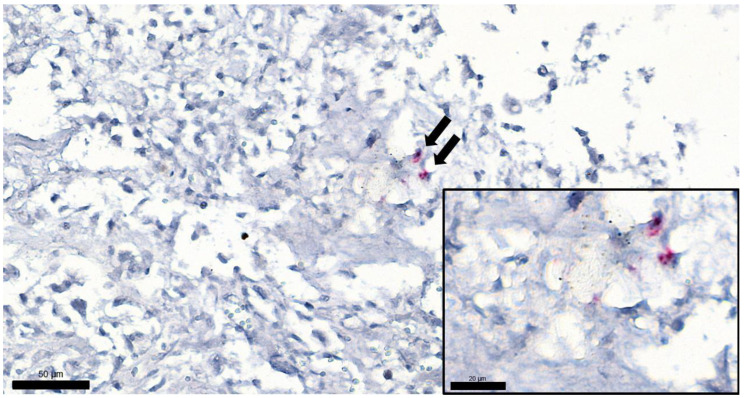
Tumor sample from a cat euthanized due to osteochondromatosis, FeLV ISH, magnification 400×, bar = 50 µm. Arrows show two positively labeled cells (presumably osteoblasts) in the tumor. Inset: same cells with higher magnification: red chromogen precipitates can be seen, showing the presence of FeLV RNA. (Magnification 1000×, bar = 20 µm).

## Data Availability

Not applicable.

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
