# Peer review of "In Situ Hybridization of Feline Leukemia Virus in a Case of Osteochondromatosis"

_vetsci, 2022, doi:10.3390/vetsci9020059_

Round 1
Reviewer 1 Report
In this study, Szilasi et al. displayes the FeLV genome within the osteochondromatosis proliferative lesions by the use of RNAscope in situ hybridization.
The evidence about the role of FeLV infections on Osteochondromatosis development and progression is still poor. Regarding this aspect, the study is interest to permit the development of new detection technique, to permit an increase of patients life quality in the future.
According to me, the manuscript can be considered acceptable for the publications with a minor revision.
In particular, to further improve the quality of the manuscript, the authors should discuss better the following points:
- It strongly recommends a checked of English language and stile of the manuscript;
- It suggests specifying in material and methods section the reagents used during the tissue digestion and etc;
- To modify the quality of images in figures 3 and 4 and the relatives’ dimension of the text;
- Improve the discussion part, for example reporting the recent evidence about the involvement of FeLV infections on Osteochondromatosis development. Regarding this, a recent paper displayed the case in which Osteochondromatosis development is independent from FeLV infections. In my opinion,this study should be mentioned in the following case report.
Author Response
Thank you for the valuable comments on our manuscript, we appreciate a lot the efforts you made.
"It strongly recommends a checked of English language and style of the manuscript." - We have made corrections regarding the style, misspellings, and English language in general, thank you for the suggestion.
"It suggests specifying in material and methods section the reagents used during the tissue digestion and etc." - We have provided more detailed information on the materials used in this case (eg. formic acid solution). Only the reagents of RNAscope 2.5 high definition [HD] red kit [from hydrogen peroxide to Gill II hematoxylin] were not specified further, as they are provided by ACDBio in a ready to use form.
"To modify the quality of images in figures 3 and 4 and the relatives’ dimension of the text." - We have made the requested changes. Additionally, figure 4D was changed to a picture of the lymph node with FeLV ISH, and the FeLV ISH result of the tumor was added as a separate figure (Fig 5) with an inset of higher magnification of positively labeled cells.
"Improve the discussion part, for example reporting the recent evidence about the involvement of FeLV infections on Osteochondromatosis development. Regarding this, a recent paper displayed the case in which Osteochondromatosis development is independent from FeLV infections. In my opinion,this study should be mentioned in the following case report." - We have improved the discussion part with more details on RNAscope method and our results. We have added the information on FeLV-negative osteochondromatosis cases (references 5, 6), but these were already cited in the earlier version of the manuscript. We could not find another, recent paper on osteochondromatosis independent from FeLV infection, however, we would definitely want to add this to our paper as you have suggested. Could you please specify which paper you think of so we can cite it properly?
Reviewer 2 Report
Dear authors,
the "case report" entitled "In situ hybridization of Feline Leukemia Virus in a case of osteochondromatosis" is very interesting and very well described.
In my personal point of view I have really appreciated the effort to well describe and study a single case report that is so important to the daily clinical practice.
I only have a curiosity and a suggestion regarding the paper.
The curiosity concerns the fact that you mentioned that the patient had a serum biochemical profile suggestive of renal failure. Did you confirm the renal damage by histology? in yes please add it into the manuscript,
The suggestion: I have work a lot with bone tumors using a decalcificator called Osteodec (Bioptica) that did not altered my immunohistochemical reactions. In the future you should try with that.
Author Response
Thank you for the valuable comments on our manuscript, we appreciate a lot the efforts you made.
"The curiosity concerns the fact that you mentioned that the patient had a serum biochemical profile suggestive of renal failure. Did you confirm the renal damage by histology? in yes please add it into the manuscript." - We did not find any alterations in renal tissue samples with histology, and we added this information to the manuscript. Probably there was a decreased glomerular filtration rate in the background of elevated urea and creatinine levels, and an SDMA level measurement would be useful to see better the renal status of this cat (or at least another blood work done after some infusion therapy). However, this could not be done due to the progressed osteochondromatosis and euthanasia.
"The suggestion: I have work a lot with bone tumors using a decalcificator called Osteodec (Bioptica) that did not altered my immunohistochemical reactions. In the future you should try with that." - Thank you very much for the suggestion, we will definitely try this product in the future.
Reviewer 3 Report
The report describes an interesting case of osteochondromatosis in a FelV-positive cat, a poorly described disease in the available literature. The novel RNAscope in situ hybridization technique was used to visualize viral RNA in osteochondromatous lesions and in other organs (spleen, lymph node, liver, brain). This is the first report where the technique is used for demonstrating FelV RNA presence osteochondromatous lesions, supporting the causative role of FelV infection, and in other tissues. The case is well documented, from radiographic appearance, to gross, histologic and RNAscope images. However, RNAscope technique seems not to be optimal for demonstrating FelV infection in osteochondromatosis lesions, since decalcifying treatment may have a major impact on RNA conservation.
Major issues
The RNAscope positive dots in the image of osteochondromatous lesion (Figure 4.D) are not clear. It would be appropriate to include a more detailed image of the positivity and/or a higher magnification image of the positive cells (e.g. an insert with cellular detail).
The novelty of the paper mainly concerns the technique and the visualization of positive cells on osteochondromatous lesions. Therefore, it would be appropriate to add in the introduction a brief explication of the RNAscope technique and a summary (and references) of the techniques already used for “in situ” visualization of FelV infection (IHC, ISH, electron microscopy and RNAscope- in the only report published by your group – Reference 16) although in other tissues. I also suggest adding a comment in discussion about the advantages and differences of RNAscope in comparison to other techniques (i.e. visualization and localization of viral infection in tissue context; higher sensitivity compared to classical ISH, visualization of viral RNA and potential for pathogenesis studies, high specificity in comparison to electron microscopy…).
This is the first case report demonstrating FelV RNA presence by RNAscope in various organs, where results are very clear and specific (spleen, brain, lymph node, liver) and this finding should be highlighted. Also, you should include results about other major organs (bone marrow, lung, kidney..) or state if they have not been sampled. It would be interesting to add more accurate descriptions of positivity in various tissues, describing the localization, distribution of positivity and the cell type infected (if possible, for example lymphocytes/macrophages, Kupffer cells/hepatocytes, glia etc.) to add a photo of lymph node. I suggest to highlight these findings also in the discussion and to mention them in the abstract.
Minor issues
Abstract
Line 12
“Viral infections mostly are considered to be the causative agents. Therefore, detection of these viral particles has become a prerequisite for diagnosing the disease.”
As you specify in the introduction, the etiology of osteochondromatosis is still not completely understood. This should also be mentioned in the abstract; for example “although the pathogenesis of osteochondromatosis is not fully understood, it is considered to be related to infection by feline leukemia virus (FeLV) or other retroviruses, as feline sarcoma virus. However, the presence of viral particles within tumor lesions has only been demonstrated by electron microscopy.”
Line 15
“The viral component attributes to the neoplastic transformation of the affected cells, most typically into osteosarcoma or chondrosarcoma.”
The sentence is not clear, please try to reformulate it. For example: “the malignant transformation of osteochondromatous lesions, most typically to osteosarcoma or chondrosarcoma, has also been attributed to the viral infection”.
Line 20
”The virus was visualized in representative tissues and in the tumorous lesions with RNAscope in situ hybridization, which confirms more the retrovirus-associated kind of this condition.”
Please use more accurate terms and form (i.e. “Viral RNA”, “tumor lesions”, “supports that Felv infection may be involved in the pathogenesis of osteochondromatosis”) and, as suggested before, highlight RNAscope results in other tissues.
Introduction
Line 36
Moreover – with a few exceptions [10–12] – feline osteochondromatosis is typically found in cats infected with feline leukemia virus (FeLV).
Please add references.
Line 38
“The presence of viral particles was observed by electron microscopy in the proliferating cells [1], but also other feline retroviruses, such as feline sarcoma virus are mentioned in the literature [13].”
The sentence is not clear, please reformulate it. For example: “Retroviral particles resembling FeLV and feline sarcoma virus has been observed by electron microscopy in the cartilage caps of cats with osteochondromatosis [1-13].
Line 41
“The term osteochondromatosis refers to the nature of the disease: multiple exostoses formed of cartilage and subchondral bone are developing,….”
The general description of the disease has been already mentioned above: I suggest deleting it.
“…..in contrast to dogs, where the disease is a developmental disturbance before skeletal maturity [14].”
I suggest moving this concept to the paragraph where differences among species are mentioned (line 33).
Line 45
“Feline osteochondromatosis has a biological behavior that is similar to one of a tumors: the exostoses show progressive growth and can substantially deteriorate the quality of life of the affected patients. Nevertheless, the tumors can become aggressive, and malignant transformation can occur towards osteosarcoma or chondrosarcoma.”
Please reformulate the sentences, since the disease is “per se” considered a neoplastic lesion and add references. For example, “Feline osteochondromatosis lesions show progressive growth and can substantially deteriorate the quality of life of the affected patients. Moreover, malignant transformation to osteosarcoma or chondrosarcoma may occur.”
Line 53
“The reported survival time is usually less than a year after the onset [11].” Change to “The reported survival time is usually less than a year after diagnosis [11].”
Line 55
The pathogenesis of feline osteochondromatosis is not fully understood, and the presence of FeLV genome in the lesions has only been confirmed by polymerase chain reaction (PCR).
Please add references.
Line 57
As viral particles almost always can be found in these lesions, the viral etiology is highly plausible, although not yet fully confirmed.
Please add references. What do you mean for “Viral particles?”. Viral infection “in sutu” has been only demonstrated by electron microscopy in sporadic reports. It could be said, more generically, that “the condition is generally associated with FelV infection”.
Line 60
The primary aim of our study was to visualize FeLV RNA in the tumor cells and other organs of a cat affected by osteochondromatosis by the use of the novel RNAscope in situ hybridization (ISH) technique we have developed for the detection of the virus [16]
You should include the description of the case as a “primary aim”.
The RNAscope technique, property of ACDBio, is usually highly standardized. Please consider deleting “we have developed” or change to “we have applied”.
Materials and methods
Line 115
“Steps of ISH were performed according to the manufacturer’s instructions (RNAscope 2.5 high definition [HD] red kit; Advanced Cell Diagnostics, Newark, CA) as described earlier [16].”
Please specify if the protocol have been applied manually or in an automatic stainer.
Please specify
Results
In case of availability, it would be interesting to include an image on the clinical aspect of the masses.
Line 142
“Every tissue sample was positive for FeLV proviral DNA with the PCR method. The RNAscope in situ hybridization also verified the presence of FeLV RNA in every tissue analyzed.”
Please list all sampled organs (i.e. the bone marrow, kidney, lung have been examined by histopathology, RNAscope and PCR?)
Line 65
“3,5-year-old castrated male European shorthair cat was admitted to a veterinary clinic, where a small (1 cm in diameter), firm nodule was noticed behind the scapula during a complete general physical examination.”
Could you specify the side? Left or right scapula?
Line 67
“The nodule was clearly visible with radiographic imaging.”
Could you synthetically describe the radiographic appearance? (i.e. radiodensity, demarcation).
Line 68
“After eight months, the first limbs’ range of motion was mildly reduced as several other nodules grew at the thoracic vertebral and sternal areas, as well as on the rib cage.”
The specification “the first limb” is not clear, please change to “the right/left forelimb”
Line 78
“The known FeLV-positive status of the animal was reconfirmed by the use of Witness FeLV-FIV test (Zoetis, Parsippany-Troy Hills, NJ).”
Please add “in peripheral blood” or other sample type used.
Line 82
Please delete: “with T-61 (MSD, Kenilworth, NJ) under general anesthesia”
Line 100
“FeLV infection status of the animal was tested with two methods: PCR and RNAscope in situ hybridization.”
Change to: “The presence of Felv genetic material in sampled tissues was tested by two methods: PCR and RNAscope in situ hybridization”
Line 101
“For the end-point polymerase chain reaction, a pool of organs (mesenteric lymph node, liver, spleen), bone marrow and the proliferative lesion were examined separately.”
Please specify if you sampled fresh tissues for PCR.
Line 171
“The presence of red signal, eg. the amount of FeLV positive cells was relatively low in the lesion that can be explained by two things: (a) there are only a few FeLV target cells in the osteochondromatous tissues that somehow are still able to drive the tumor growth, or (b) more, yet to be identified cells and cell types are infected, but the formic acid solution during the process of decalcification degrades the RNA leading to loss of sensitivity.”
Please add references about consequences on RNA degradation of decalcification process.
i.e. L Walsh , A J Freemont, J A Hoyland. The effect of tissue decalcification on mRNA retention within bone for in-situ hybridization studies Int J Exp Pathol. 1993 Jun;74(3):237-41.
Line 177
Previous case reports mentioned only the use of electron microscopy, polymerase chain reaction and immunohistochemistry [1–4] to confirm FeLV infection, and, except electron microscopy, these methods were unsuccessful to detect the virus in the osteochondromas.
The sentence is not clear; moreover, in line 55 you stated “The pathogenesis of feline osteochondromatosis is not fully understood, and the presence of FeLV genome in the lesions has only been confirmed by polymerase chain reaction (PCR).” Please reformulate. For example “To the Author’s knowledge, the presence of retroviral particles in situ within tumor lesions, has been only demonstrated by electron microscopy. In one report, immunohistrochemistry failed to detect FelV antigens in tumor cells, whether positivity was observed in hematopoietic cells of bone marrow.”
Line 182
“Demonstrability of viral particles was hindered due to formic acid decalcification (and therefore destruction of FeLV RNA), as it was confirmed by the use of a housekeeping positive control probe, but this step is inevitable in processing osseus tissues.”
It could be appropriate to discuss the use of other decalcifying agents (i.e. EDTA) and to add a sentence of conclusion.
Author Response
Thank you for the valuable and very detailed comments and suggestions on our manuscript, we appreciate a lot the efforts you made to improve the quality of this paper.
We have made corrections regarding the style, misspellings, and English language in general, thank you for the suggestion.
Major issues
The RNAscope positive dots in the image of osteochondromatous lesion (Figure 4.D) are not clear. It would be appropriate to include a more detailed image of the positivity and/or a higher magnification image of the positive cells (e.g. an insert with cellular detail). - We have made the requested changes. Figure 4D was changed to a picture of the mesenteric lymph node with FeLV ISH, and the FeLV ISH result of the tumor was added as a separate figure (Fig 5) with an inset of higher magnification of positively labeled cells. We have considered the opportunity to add a panel of pictures on the tumor ISH results (displaying that not only these cells are positive), but all labeled areas look quite similar to the area chosen (figure 5), therefore we recanted this idea.
The novelty of the paper mainly concerns the technique and the visualization of positive cells on osteochondromatous lesions. Therefore, it would be appropriate to add in the introduction a brief explication of the RNAscope technique and a summary (and references) of the techniques already used for “in situ” visualization of FelV infection (IHC, ISH, electron microscopy and RNAscope- in the only report published by your group – Reference 16) although in other tissues. I also suggest adding a comment in discussion about the advantages and differences of RNAscope in comparison to other techniques (i.e. visualization and localization of viral infection in tissue context; higher sensitivity compared to classical ISH, visualization of viral RNA and potential for pathogenesis studies, high specificity in comparison to electron microscopy…). - Thank you for the suggestions. A paragraph was added to the introduction detailing in situ visualization methods and RNAscope, and also a paragraph was added to the discussion.
This is the first case report demonstrating FelV RNA presence by RNAscope in various organs, where results are very clear and specific (spleen, brain, lymph node, liver) and this finding should be highlighted. Also, you should include results about other major organs (bone marrow, lung, kidney..) or state if they have not been sampled. - We have added more information on ISH results on every organ analyzed, and stated that bone marrow was not examined with this method. It would be interesting to add more accurate descriptions of positivity in various tissues, describing the localization, distribution of positivity and the cell type infected (if possible, for example lymphocytes/macrophages, Kupffer cells/hepatocytes, glia etc.) to add a photo of lymph node. I suggest to highlight these findings also in the discussion and to mention them in the abstract. - Information has been added to the results chapter, and these were also highlighted in the abstract and discussion. A picture of mesenteric lymph node with FeLV ISH has been added to figure 4.
Minor issues
Abstract
Line 12
“Viral infections mostly are considered to be the causative agents. Therefore, detection of these viral particles has become a prerequisite for diagnosing the disease.”
As you specify in the introduction, the etiology of osteochondromatosis is still not completely understood. This should also be mentioned in the abstract; for example “although the pathogenesis of osteochondromatosis is not fully understood, it is considered to be related to infection by feline leukemia virus (FeLV) or other retroviruses, as feline sarcoma virus. However, the presence of viral particles within tumor lesions has only been demonstrated by electron microscopy.” - We changed the sentence as you have kindly suggested.
Line 15
“The viral component attributes to the neoplastic transformation of the affected cells, most typically into osteosarcoma or chondrosarcoma.”
The sentence is not clear, please try to reformulate it. For example: “the malignant transformation of osteochondromatous lesions, most typically to osteosarcoma or chondrosarcoma, has also been attributed to the viral infection”. - We changed the sentence as you have kindly suggested.
Line 20
”The virus was visualized in representative tissues and in the tumorous lesions with RNAscope in situ hybridization, which confirms more the retrovirus-associated kind of this condition.”
Please use more accurate terms and form (i.e. “Viral RNA”, “tumor lesions”, “supports that Felv infection may be involved in the pathogenesis of osteochondromatosis”) and, as suggested before, highlight RNAscope results in other tissues. - We changed the sentence as you have kindly suggested.
Introduction
Line 36
Moreover – with a few exceptions [10–12] – feline osteochondromatosis is typically found in cats infected with feline leukemia virus (FeLV).
Please add references. - References are now added.
Line 38
“The presence of viral particles was observed by electron microscopy in the proliferating cells [1], but also other feline retroviruses, such as feline sarcoma virus are mentioned in the literature [13].”
The sentence is not clear, please reformulate it. For example: “Retroviral particles resembling FeLV and feline sarcoma virus has been observed by electron microscopy in the cartilage caps of cats with osteochondromatosis [1-13]. - We changed the sentence as you have kindly suggested.
Line 41
“The term osteochondromatosis refers to the nature of the disease: multiple exostoses formed of cartilage and subchondral bone are developing,….”
The general description of the disease has been already mentioned above: I suggest deleting it. - We have deleted it.
“…..in contrast to dogs, where the disease is a developmental disturbance before skeletal maturity [14].”
I suggest moving this concept to the paragraph where differences among species are mentioned (line 33). - This was moved to an earlier part of the introduction as you have suggested.
Line 45
“Feline osteochondromatosis has a biological behavior that is similar to one of a tumors: the exostoses show progressive growth and can substantially deteriorate the quality of life of the affected patients. Nevertheless, the tumors can become aggressive, and malignant transformation can occur towards osteosarcoma or chondrosarcoma.”
Please reformulate the sentences, since the disease is “per se” considered a neoplastic lesion and add references. For example, “Feline osteochondromatosis lesions show progressive growth and can substantially deteriorate the quality of life of the affected patients. Moreover, malignant transformation to osteosarcoma or chondrosarcoma may occur.” - We changed the sentence as you have kindly suggested.
Line 53
“The reported survival time is usually less than a year after the onset [11].” Change to “The reported survival time is usually less than a year after diagnosis [11].” - We changed the sentence as you have kindly suggested.
Line 55
The pathogenesis of feline osteochondromatosis is not fully understood, and the presence of FeLV genome in the lesions has only been confirmed by polymerase chain reaction (PCR).
Please add references. - References are now added.
Line 57
As viral particles almost always can be found in these lesions, the viral etiology is highly plausible, although not yet fully confirmed.
Please add references. What do you mean for “Viral particles?”. Viral infection “in sutu” has been only demonstrated by electron microscopy in sporadic reports. It could be said, more generically, that “the condition is generally associated with FelV infection”. - We changed the sentence as you have kindly suggested, and provided references.
Line 60
The primary aim of our study was to visualize FeLV RNA in the tumor cells and other organs of a cat affected by osteochondromatosis by the use of the novel RNAscope in situ hybridization (ISH) technique we have developed for the detection of the virus [16]
You should include the description of the case as a “primary aim”. - We changed the sentence as you have kindly suggested.
The RNAscope technique, property of ACDBio, is usually highly standardized. Please consider deleting “we have developed” or change to “we have applied”. - We have deleted this sentence.
Materials and methods
Line 115
“Steps of ISH were performed according to the manufacturer’s instructions (RNAscope 2.5 high definition [HD] red kit; Advanced Cell Diagnostics, Newark, CA) as described earlier [16].”
Please specify if the protocol have been applied manually or in an automatic stainer. - It was done manually, now it is in the manuscript.
Please specify
Results
In case of availability, it would be interesting to include an image on the clinical aspect of the masses. - Unfortunately, we don’t have any pictures of the animal before the necropsy.
Line 142
“Every tissue sample was positive for FeLV proviral DNA with the PCR method. The RNAscope in situ hybridization also verified the presence of FeLV RNA in every tissue analyzed.”
Please list all sampled organs (i.e. the bone marrow, kidney, lung have been examined by histopathology, RNAscope and PCR?) - Now the detailed lists are available in the text where we mention a method.
Line 65
“3,5-year-old castrated male European shorthair cat was admitted to a veterinary clinic, where a small (1 cm in diameter), firm nodule was noticed behind the scapula during a complete general physical examination.”
Could you specify the side? Left or right scapula? - Left scapula, added to the manuscript.
Line 67
“The nodule was clearly visible with radiographic imaging.”
Could you synthetically describe the radiographic appearance? (i.e. radiodensity, demarcation). - "The nodule was clearly visible with radiographic imaging – a calcified, radiodense, and well-circumscribed lesion could be seen."
Line 68
“After eight months, the first limbs’ range of motion was mildly reduced as several other nodules grew at the thoracic vertebral and sternal areas, as well as on the rib cage.”
The specification “the first limb” is not clear, please change to “the right/left forelimb” - „both forelimbs” was added to the text.
Line 78
“The known FeLV-positive status of the animal was reconfirmed by the use of Witness FeLV-FIV test (Zoetis, Parsippany-Troy Hills, NJ).”
Please add “in peripheral blood” or other sample type used. - „in peripheral blood” was added to this sentence.
Line 82
Please delete: “with T-61 (MSD, Kenilworth, NJ) under general anesthesia” - It is now deleted.
Line 100
“FeLV infection status of the animal was tested with two methods: PCR and RNAscope in situ hybridization.”
Change to: “The presence of Felv genetic material in sampled tissues was tested by two methods: PCR and RNAscope in situ hybridization” - We changed the sentence as you have kindly suggested.
Line 101
“For the end-point polymerase chain reaction, a pool of organs (mesenteric lymph node, liver, spleen), bone marrow and the proliferative lesion were examined separately.”
Please specify if you sampled fresh tissues for PCR. - Fresh tissues were used for PCR, now it is added to the text.
Line 171
“The presence of red signal, eg. the amount of FeLV positive cells was relatively low in the lesion that can be explained by two things: (a) there are only a few FeLV target cells in the osteochondromatous tissues that somehow are still able to drive the tumor growth, or (b) more, yet to be identified cells and cell types are infected, but the formic acid solution during the process of decalcification degrades the RNA leading to loss of sensitivity.”
Please add references about consequences on RNA degradation of decalcification process.
i.e. L Walsh , A J Freemont, J A Hoyland. The effect of tissue decalcification on mRNA retention within bone for in-situ hybridization studies Int J Exp Pathol. 1993 Jun;74(3):237-41. - Thank you for the suggested reference, it was added to the manuscript.
Line 177
Previous case reports mentioned only the use of electron microscopy, polymerase chain reaction and immunohistochemistry [1–4] to confirm FeLV infection, and, except electron microscopy, these methods were unsuccessful to detect the virus in the osteochondromas.
The sentence is not clear; moreover, in line 55 you stated “The pathogenesis of feline osteochondromatosis is not fully understood, and the presence of FeLV genome in the lesions has only been confirmed by polymerase chain reaction (PCR).” Please reformulate. For example “To the Author’s knowledge, the presence of retroviral particles in situ within tumor lesions, has been only demonstrated by electron microscopy. In one report, immunohistrochemistry failed to detect FelV antigens in tumor cells, whether positivity was observed in hematopoietic cells of bone marrow.” - We changed the sentence as you have kindly suggested.
Line 182
“Demonstrability of viral particles was hindered due to formic acid decalcification (and therefore destruction of FeLV RNA), as it was confirmed by the use of a housekeeping positive control probe, but this step is inevitable in processing osseus tissues.”
It could be appropriate to discuss the use of other decalcifying agents (i.e. EDTA) and to add a sentence of conclusion. - We have added some sentences to the discussion regarding decalcification, and also a sentence to the end, as a conclusion.
Round 2
Reviewer 3 Report
Authors have significantly improved the general arrangement of the manuscript.
The insert reporting the higher magnification of RNAscope positivity within lesions and the image of lymph node are very clear and useful.
I suggest other minor revisions that the authors might consider.
Line 17
“Therefore, detection of these viral particles has become a prerequisite for diagnosing the disease.”
Viral particles have only sporadically been detected in lesions and the diagnosis of osteochondromatosis is based on clinical and anatomo-histopathological findings. I suggest deleting the sentence.
Line 21
It affects the axial skeleton: the most common sites in decreasing frequency of occurrence being are the rib cage, scapulae, vertebral column, skull, pelvis, and limbs.
The information sounds redundant in the abstract. I suggest deleting the sentence.
Line 40
"Besides cats, the disease has been reported in pigs, horses, dogs, and a macaque."
Please consider adding “Besides cats, the disease has been reported in horses, dogs, pigs, and a macaque. The syndrome has shown a familial/hereditary background in horses, dogs and pigs, similarly to humans, and it occurs typically in young animals during the period of active bone growth.” Reference: Chester, D.K. (1971) Multiple cartilaginous exostoses in two generations of dogs. J Am Vet Med Assoc 159:895–897.)
Line 42
“Compared to these species where the condition has a familial /hereditary background [12], feline osteochondromatosis has a different etiology, pathogenesis, and overall biology:
Consider changing to “Feline osteochondromatosis shows different pathobiology, clinical aspect, and most likely has a different etiology: it is predominantly seen in young cats, where it is most often diagnosed at the age of 2–4 years [2], when the skeleton has already matured.”
Line 60
Moreover, malignant transformation to osteosarcoma or chondrosarcoma may occur.
Please add references.
In the abstract (line 18) you also stated that malignant transformation is considered virus-related (The malignant transformation of osteochondromatous lesions, most typically to osteosarcoma or chondrosarcoma, has also been attributed to the viral infection): please verify this information, I could not find references.
Line 72 -79
The paragraph from “The pathogenesis” to “other species” continues the argument addressed before, in lines 42-50.
I suggest to summarize this part as, for example, “However, the role of FelV infection in the development of this disease have not been fully elucidated. The viral etiology might explain the different manifestation of the disease in cats compared to other species.” And to move at the end of the paragraph of lines 42-50. Moreover references 1,4 of the sentence “The pathogenesis of feline osteochondromatosis is not fully understood, and the presence of FeLV genome in the lesions in some cases has only been confirmed by polymerase chain reaction (PCR) and electron microscopy [1, 4].” do not reported PCR positivity for felv of osteochondromatous lesions but only of peripheral blood. Therefore, I suggest adding correct references or to delete the sentence.
Line 81
"Several techniques have been already used for “in situ” visualization of FeLV in 81 tissues in general (not in osteochondromatosis), such as immunodiffusion, electron microscopy, immunohistochemistry, and fluorescent in situ hybridization [17–19]."
To my knowledge, immunodiffusion technique do not consent visualization of infection in tissues. I suggest changing to: “Several techniques have been already used for the visualization of FeLV infection in tissue sections, such as electron microscopy, immunohistochemistry, and fluorescent in situ hybridization”
Line 84
“RNAscope in situ hybridization (ISH) is a novel technique based on the detection of messenger RNA molecules in tissue samples [20].”
RNA scope can also detect viral RNA, I suggest deleting “messenger”
Line 85
This method detects actively replicating viruses (FeLV in our case) in tissues. It is more sensitive and specific than the methods mentioned above, which gives the possibility to observe virus quantity and also localization in cells. Besides, the distribution and type of virally infected cells can be examined 88 with this method more specifically, than with immunohistochemistry.
I think the first sentence is not completely correct, since you should also detect proviral DNA. Consider changing this part to “RNAscope has a higher sensitivity compared to classic ISH which enables the detection of low viral loads with extreme specificity. Moreover, it provides a very accurate localization and visualization of infected cell type.”
Line 90
"The primary aim of our study was to describe the case of a 3,5-year-old European shorthair cat affected by osteochondromatosis and to visualize FeLV RNA in the tumor cells and other organs of the carcass using the novel RNAscope in situ hybridization technique [21]."
Please consider changing to: "The primary aim of our study was to describe the case of a 3,5-year-old European shorthair cat affected by osteochondromatosis and detect the presence of FeLV RNA within the tumor tissue and other organs using the novel RNAscope in situ hybridization technique [21]."
I suggest changing the order of figures: Figure 4. Image of RNAscope in osteochondroma; Figure 5 Image of RNAscope in other organs.
Line 180
"Routine histopathologic examination of sampled organs with hematoxylin and eosin staining revealed mild pathologic lesions only: mild diffuse microvesicular steatohepathopathy in the liver, multifocal nodular hyperplasia in the pancreas, and mild lymphoid depletion and paracortical hyperplasia in the mesenteric lymph node. We did not find any alterations in kidney sections, which could explain the increased urea and creatinine levels in peripheral blood. Histopathologic examination of the representative samples obtained from the bony masses revealed multiple irregular islands of well-differentiated hyaline cartilage surrounded by osteoid tissue and bone trabeculae originating from endochondral ossification (Figure 3)."
I suggest deleting occasional findings as nodular hyperplasia of pancreas and the all sentence on the kidney and moving these minor at the end of the paragraph, after the description of bony masses.
Line 196
“The RNAscope in situ hybridization also verified the presence of FeLV RNA in every tissue analyzed (liver, spleen, kidney, mesenteric lymph node, lung, brain, osteochondroma).”
I suggest changing to “FelV RNA was detected by RNAscope in situ hybridization in osteochondromas and every other organs analyzed (liver, spleen, kidney, mesenteric lymph node, lung, brain)”
Line 225
"A 3,5-year-old castrated male domestic European shorthair cat was euthanized due to the progressive proliferation of large osseous masses connected to the sternum, ribs, and the spinal column of the animal."
I suggest changing “castrated” to “neutered”
Line 232
Previous reports verified the presence of FeLV of the affected animals by PCR mostly [3, 4], however within the osteochondromatosis lesions the presence of retrovirus-resembling particles was only seen with electron micros copy [1, 4].
In references 3,4 felv PCR was not performed from osteocrondromatosis lesions. I suggest finding accurate references or deleting this part of the sentence.
Line 232
I suggest adding in the discussion a sentence about RNAscope positivity on other organs without histopathological lesions.
Line 260
Demonstrability of the viral genome was probably hindered due to formic acid decalcification (and therefore destruction of FeLV RNA), as it was confirmed by the use of a housekeeping positive control probe [23], but this step is inevitable in processing osseous tissues.
I suggest changing “destruction” to “degradation”
Line 264
"It would be advisable to examine every feline osteochondromatosis case in the future with RNAscope ISH assay to confirm or deny the presence of FeLV RNA in the tumorous lesions, but with a less destructive decalcification method."
This sencence is redundant, I suggest deleting it.
I suggest moving the last paragraph about decalcification process (Lines 260-267) before the paragraph of lines 250-259.
Author Response
Thank you again for the accurate revision. We have made all changes required, and also have checked for spelling mistakes.
Line 17
“Therefore, detection of these viral particles has become a prerequisite for diagnosing the disease.”
Viral particles have only sporadically been detected in lesions and the diagnosis of osteochondromatosis is based on clinical and anatomo-histopathological findings. I suggest deleting the sentence. - It is deleted.
Line 21
It affects the axial skeleton: the most common sites in decreasing frequency of occurrence being are the rib cage, scapulae, vertebral column, skull, pelvis, and limbs.
The information sounds redundant in the abstract. I suggest deleting the sentence. - It is deleted.
Line 40
"Besides cats, the disease has been reported in pigs, horses, dogs, and a macaque."
Please consider adding “Besides cats, the disease has been reported in horses, dogs, pigs, and a macaque. The syndrome has shown a familial/hereditary background in horses, dogs and pigs, similarly to humans, and it occurs typically in young animals during the period of active bone growth.” Reference: Chester, D.K. (1971) Multiple cartilaginous exostoses in two generations of dogs. J Am Vet Med Assoc 159:895–897.) - Sentence was changed accordingly, and this reference was added (ref 12)
Line 42
“Compared to these species where the condition has a familial /hereditary background [12], feline osteochondromatosis has a different etiology, pathogenesis, and overall biology:
Consider changing to “Feline osteochondromatosis shows different pathobiology, clinical aspect, and most likely has a different etiology: it is predominantly seen in young cats, where it is most often diagnosed at the age of 2–4 years [2], when the skeleton has already matured.”- Sentence was changed accordingly.
Line 60
Moreover, malignant transformation to osteosarcoma or chondrosarcoma may occur.
Please add references.
In the abstract (line 18) you also stated that malignant transformation is considered virus-related (The malignant transformation of osteochondromatous lesions, most typically to osteosarcoma or chondrosarcoma, has also been attributed to the viral infection): please verify this information, I could not find references. - ref 17 was added to cover this information
Line 72 -79
The paragraph from “The pathogenesis” to “other species” continues the argument addressed before, in lines 42-50.
I suggest to summarize this part as, for example, “However, the role of FelV infection in the development of this disease have not been fully elucidated. The viral etiology might explain the different manifestation of the disease in cats compared to other species.” And to move at the end of the paragraph of lines 42-50. Moreover references 1,4 of the sentence “The pathogenesis of feline osteochondromatosis is not fully understood, and the presence of FeLV genome in the lesions in some cases has only been confirmed by polymerase chain reaction (PCR) and electron microscopy [1, 4].” do not reported PCR positivity for felv of osteochondromatous lesions but only of peripheral blood. Therefore, I suggest adding correct references or to delete the sentence.- Sentences was changed accordingly, moved to the end of desired paragraph. For the PCR detection, sentence was changed to "The pathogenesis of feline osteochondromatosis is not fully understood, and the concurrent FeLV infection in some cases has only been confirmed from peripheral blood by polymerase chain reaction (PCR)."
Line 81
"Several techniques have been already used for “in situ” visualization of FeLV in 81 tissues in general (not in osteochondromatosis), such as immunodiffusion, electron microscopy, immunohistochemistry, and fluorescent in situ hybridization [17–19]."
To my knowledge, immunodiffusion technique do not consent visualization of infection in tissues. I suggest changing to: “Several techniques have been already used for the visualization of FeLV infection in tissue sections, such as electron microscopy, immunohistochemistry, and fluorescent in situ hybridization” - Sentence was changed accordingly, thank you for clarification.
Line 84
“RNAscope in situ hybridization (ISH) is a novel technique based on the detection of messenger RNA molecules in tissue samples [20].”
RNA scope can also detect viral RNA, I suggest deleting “messenger” - Deleted.
Line 85
This method detects actively replicating viruses (FeLV in our case) in tissues. It is more sensitive and specific than the methods mentioned above, which gives the possibility to observe virus quantity and also localization in cells. Besides, the distribution and type of virally infected cells can be examined 88 with this method more specifically, than with immunohistochemistry.
I think the first sentence is not completely correct, since you should also detect proviral DNA. Consider changing this part to “RNAscope has a higher sensitivity compared to classic ISH which enables the detection of low viral loads with extreme specificity. Moreover, it provides a very accurate localization and visualization of infected cell type.” - Sentence was changed accordingly.
Line 90
"The primary aim of our study was to describe the case of a 3,5-year-old European shorthair cat affected by osteochondromatosis and to visualize FeLV RNA in the tumor cells and other organs of the carcass using the novel RNAscope in situ hybridization technique [21]."
Please consider changing to: "The primary aim of our study was to describe the case of a 3,5-year-old European shorthair cat affected by osteochondromatosis and detect the presence of FeLV RNA within the tumor tissue and other organs using the novel RNAscope in situ hybridization technique [21]." - Sentence was changed accordingly.
I suggest changing the order of figures: Figure 4. Image of RNAscope in osteochondroma; Figure 5 Image of RNAscope in other organs.
Order of figures 4 and 5 was changed.
Line 180
"Routine histopathologic examination of sampled organs with hematoxylin and eosin staining revealed mild pathologic lesions only: mild diffuse microvesicular steatohepathopathy in the liver, multifocal nodular hyperplasia in the pancreas, and mild lymphoid depletion and paracortical hyperplasia in the mesenteric lymph node. We did not find any alterations in kidney sections, which could explain the increased urea and creatinine levels in peripheral blood. Histopathologic examination of the representative samples obtained from the bony masses revealed multiple irregular islands of well-differentiated hyaline cartilage surrounded by osteoid tissue and bone trabeculae originating from endochondral ossification (Figure 3)."
I suggest deleting occasional findings as nodular hyperplasia of pancreas and the all sentence on the kidney and moving these minor at the end of the paragraph, after the description of bony masses. - Sentence was changed accordingly, moved to the end of paragraph. Reviewer 1 was asking for detailed and specific description of histopathologic results of other organs, which we found also a good idea, therefore we would like to keep pancreas results also, if this is possible.
Line 196
“The RNAscope in situ hybridization also verified the presence of FeLV RNA in every tissue analyzed (liver, spleen, kidney, mesenteric lymph node, lung, brain, osteochondroma).”
I suggest changing to “FelV RNA was detected by RNAscope in situ hybridization in osteochondromas and every other organs analyzed (liver, spleen, kidney, mesenteric lymph node, lung, brain)”- Sentence was changed accordingly.
Line 225
"A 3,5-year-old castrated male domestic European shorthair cat was euthanized due to the progressive proliferation of large osseous masses connected to the sternum, ribs, and the spinal column of the animal."
I suggest changing “castrated” to “neutered” - It was changed to neutered.
Line 232
Previous reports verified the presence of FeLV of the affected animals by PCR mostly [3, 4], however within the osteochondromatosis lesions the presence of retrovirus-resembling particles was only seen with electron micros copy [1, 4].
In references 3,4 felv PCR was not performed from osteocrondromatosis lesions. I suggest finding accurate references or deleting this part of the sentence. - Sentence was reformed, so it is visible that PCR tests were done only on peripheral blood samples. "Previous reports verified the presence of FeLV of the affected animals in peripheral blood by PCR mostly [3, 4], however within the osteochondromatosis lesions the presence of retrovirus-resembling particles was only seen with electron microscopy [1, 4]."
Line 232
I suggest adding in the discussion a sentence about RNAscope positivity on other organs without histopathological lesions. - A sentence was added "Moreover, we could detect the virus in tissues without severe histopathologic lesions, such as liver, spleen, kidney, mesenteric lymph node, lung, and brain." to lines 203-204.
Line 260
Demonstrability of the viral genome was probably hindered due to formic acid decalcification (and therefore destruction of FeLV RNA), as it was confirmed by the use of a housekeeping positive control probe [23], but this step is inevitable in processing osseous tissues.
I suggest changing “destruction” to “degradation” - Changed to degradation.
Line 264
"It would be advisable to examine every feline osteochondromatosis case in the future with RNAscope ISH assay to confirm or deny the presence of FeLV RNA in the tumorous lesions, but with a less destructive decalcification method."
This sencence is redundant, I suggest deleting it. - It is deleted.
I suggest moving the last paragraph about decalcification process (Lines 260-267) before the paragraph of lines 250-259. - The paragraph has been moved to the suggested place.